# Attention U-Net:
# Learning Where to Look for the Pancreas

Ozan Oktay[1,5], Jo Schlemper[1], Loic Le Folgoc[1], Matthew Lee[4], Mattias Heinrich[3],
Kazunari Misawa[2], Kensaku Mori[2], Steven McDonagh[1], Nils Y Hammerla[5],
Bernhard Kainz[1], Ben Glocker[1], and Daniel Rueckert[1]

[1]Biomedical Image Analysis Group, Imperial College London, London, UK
[2]Dept. of Media Science, Nagoya University & Aichi Cancer Center, JP
[3]Medical Informatics, University of Luebeck, DE, [4]HeartFlow, California, USA
[5]Babylon Health, London, UK

## Abstract

We propose a novel attention gate (AG) model for medical imaging that automatically learns to focus on target structures of varying shapes and sizes. Models trained with AGs implicitly learn to suppress irrelevant regions in an input image while highlighting salient features useful for a specific task. This enables us to eliminate the necessity of using explicit external tissue/organ localisation modules of cascaded convolutional neural networks (CNNs). AGs can be easily integrated into standard CNN architectures such as the U-Net model with minimal computational overhead while increasing the model sensitivity and prediction accuracy. The proposed Attention U-Net architecture is evaluated on two large CT abdominal datasets for multi-class image segmentation. Experimental results show that AGs consistently improve the prediction performance of U-Net across different datasets and training sizes while preserving computational efficiency. The source code for the proposed architecture is publicly available.

## 1 Introduction

Automated medical image segmentation has been extensively studied in the image analysis community due to the fact that manual, dense labelling of large amounts of medical images is a tedious and error-prone task. Accurate and reliable solutions are desired to increase clinical work flow efficiency and support decision making through fast and automatic extraction of quantitative measurements.

With the advent of convolutional neural networks (CNNs), near-radiologist level performance can be achieved in automated medical image analysis tasks including cardiac MR segmentation [3] and cancerous lung nodule detection [17]. High representation power, fast inference, and filter sharing properties have made CNNs the de facto standard for image segmentation. Fully convolutional networks (FCNs) [18] and the U-Net [24] are two commonly used architectures. Despite their good representational power, these architectures rely on multi-stage cascaded CNNs when the target organs show large inter-patient variation in terms of shape and size. Cascaded frameworks extract a region of interest (ROI) and make dense predictions on that particular ROI. The application areas include cardiac MRI [14], cardiac CT [23], abdominal CT [26, 27] segmentation, and lung CT nodule detection [17]. However, this approach leads to excessive and redundant use of computational resources and model parameters; for instance, similar low-level features are repeatedly extracted by all models within the cascade. To address this general problem, we propose a simple and yet effective solution, namely *attention gates* (AGs). CNN models with AGs can be trained from scratch in a standard way similar to the training of a FCN model, and AGs automatically learn to focus on target

1st Conference on Medical Imaging with Deep Learning (MIDL 2018), Amsterdam, The Netherlands.

structures without additional supervision. At test time, these gates generate soft region proposals implicitly on-the-fly and highlight salient features useful for a specific task. Moreover, they do not introduce significant computational overhead and do not require a large number of model parameters as in the case of multi-model frameworks. In return, the proposed AGs improve model sensitivity and accuracy for dense label predictions by suppressing feature activations in irrelevant regions. In this way, the necessity of using an external organ localisation model can be eliminated while maintaining the high prediction accuracy. Similar attention mechanisms have been proposed for natural image classification [11] and captioning [1] to perform adaptive feature pooling, where model predictions are conditioned only on a subset of selected image regions. In this paper, we generalise this design and propose image-grid based gating that allows attention coefficients to be specific to local regions. Moreover, our approach can be used for attention-based dense predictions.

We demonstrate the implementation of AG in a standard U-Net architecture (*Attention U-Net*) and apply it to medical images. We choose the challenging CT pancreas segmentation problem to provide experimental evidence for our proposed contributions. This problem constitutes a difficult task due to low tissue contrast and large variability in organ shape and size. We evaluate our implementation on two commonly used benchmarks: TCIA Pancreas $CT$-82 [25] and multi-class abdominal $CT$-150. The results show that AGs consistenly improve prediction accuracy across different datasets and training sizes while achieving state-of-the-art performance without requiring multiple CNN models.

## 1.1 Related Work

**CT Pancreas Segmentation:** Early work on pancreas segmentation from abdominal CT used statistical shape models [5, 28] or multi-atlas techniques [22, 34]. In particular, atlas approaches benefit from implicit shape constraints enforced by propagation of manual annotations. However, in public benchmarks such as the TCIA dataset [25], Dice similarity coefficients (DSC) for atlas-based frameworks ranges from $69.6\%$ to $73.9\%$ [22, 34]. In [39] a classification based framework is proposed to remove the dependency of atlas to image registration. Recently, cascaded multi-stage CNN models [26, 27, 38] have been proposed to address the problem. Here, an initial coarse-level model (e.g. U-Net or Regression Forest) is used to obtain a ROI and then a cropped ROI is used for segmentation refinement by a second model. Similarly, combinations of 2D-FCN and recurrent neural network (RNN) models are utilised in [4] to exploit dependencies between adjacent axial slices. These approaches achieve state-of-the-art performance in the TCIA benchmark ($81.2\% - 82.4\%$ DSC). Without using a cascaded framework, the performance drops between $2.0\%$ and $4.4\%$. Recent work [37] proposed an iterative two-stage model that recursively updates local and global predictions, and both models are trained end-to-end. Besides standard FCNs, dense connections [6] and sparse convolutions [8, 9] have been applied to the CT pancreas segmentation problem. Dense connections and sparse kernels reduce computational complexity by requiring less number of non-zero parameters.

**Attention Gates:** AGs are commonly used in natural image analysis, knowledge graphs, and language processing (NLP) for image captioning [1], machine translation [2, 30], and classification [11, 31, 32] tasks. Initial work has explored attention-maps by interpreting gradient of output class scores with respect to the input image. Trainable attention, on the other hand, is enforced by design and categorised as hard- and soft-attention. Hard attention [21], e.g. iterative region proposal and cropping, is often non-differentiable and relies on reinforcement learning for parameter updates, which makes model training more difficult. Recursive hard-attention is used in [36] to detect anomalies in chest X-ray scans. Contrarily, soft attention is probabilistic and utilises standard back-propagation without need for Monte Carlo sampling. For instance, additive soft attention is used in sentence-to-sentence translation [2, 29] and more recently applied to image classification [11, 32]. In [10], channel-wise attention is used to highlight important feature dimensions, which was the top-performer in the ILSVRC 2017 image classification challenge. Self-attention techniques [11, 33] have been proposed to remove the dependency on external gating information. For instance, non-local self attention is used in [33] to capture long range dependencies. In [11, 32] self-attention is used to perform class-specific pooling, which results in more accurate and robust image classification performance.

## 1.2 Contributions

In this paper, we propose a novel self-attention gating module that can be utilised in CNN based standard image analysis models for dense label predictions. Moreover, we explore the benefit of AGs to medical image analysis, in particular, in the context of image segmentation. The contributions of this work can be summarised as follows:

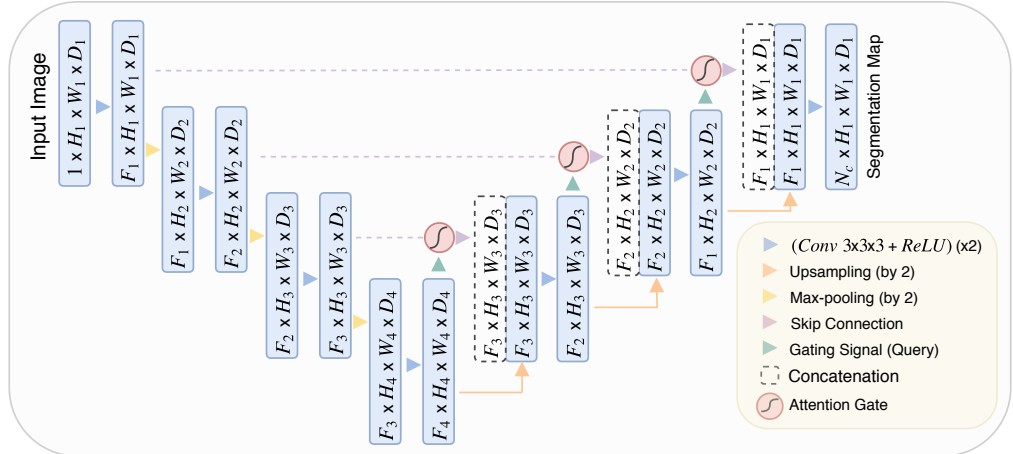

Figure 1: A block diagram of the proposed Attention U-Net segmentation model. Input image is progressively filtered and downsampled by factor of 2 at each scale in the encoding part of the network (e.g. $H_4 = H_1/8$). $N_c$ denotes the number of classes. Attention gates (AGs) filter the features propagated through the skip connections. Schematic of the AGs is shown in Figure 2. Feature selectivity in AGs is achieved by use of contextual information (gating) extracted in coarser scales.

- We take the attention approach proposed in [11] a step further by proposing grid-based gating that allows attention coefficients to be more specific to local regions. This improves performance compared to gating based on a global feature vector. Moreover, our approach can be used for dense predictions since we do not perform adaptive pooling.

- We propose one of the first use cases of soft-attention technique in a feed-forward CNN model applied to a medical imaging task. The proposed attention gates can replace hard-attention approaches used in image classification [36] and external organ localisation models in image segmentation frameworks [14, 22, 26, 27].

- An extension to the standard U-Net model is proposed to improve model sensitivity to foreground pixels without requiring complicated heuristics. Accuracy improvements over U-Net are experimentally observed to be consistent across different imaging datasets.

## 2   Methodology

**Fully Convolutional Network (FCN):** Convolutional neural networks (CNNs) outperform traditional approaches in medical image analysis on public benchmark datasets [14, 17] while being an order of magnitude faster than, e.g., graph-cut and multi-atlas segmentation techniques [34]. This is mainly attributed to the fact that (I) domain specific image features are learnt using stochastic gradient descent (SGD) optimisation, (II) learnt kernels are shared across all pixels, and (III) image convolution operations exploit the structural information in medical images well. In particular, fully convolutional networks (FCN) [18] such as U-Net [24], DeepMedic [13] and holistically nested networks [16, 35] have been shown to achieve robust and accurate performance in various tasks including cardiac MR [3], brain tumours [12] and abdominal CT [26, 27] image segmentation tasks.

Convolutional layers progressively extract higher dimensional image representations ($x^l$) by processing local information layer by layer. Eventually, this separates pixels in a high dimensional space according to their semantics. Through this sequential process, model predictions are conditioned on information collected from a large receptive field. Hence, feature-map $x^l$ is obtained at the output of layer $l$ by sequentially applying a linear transformation followed by a non-linear activation function. It is often chosen as rectified linear unit: $\sigma_1(x^l_{i,c}) = max(0, x^l_{i,c})$ where $i$ and $c$ denote spatial and channel dimensions respectively. Feature activations can be formulated as: $x^l_c = \sigma_1\left(\sum_{c' \in F_l} x^{l-1}_{c'} * k_{c',c}\right)$ where $*$ denotes the convolution operation, and the spatial subscript ($i$) is omitted in the formulation for notational clarity. The function $f(x^l; \Phi^l) = x^{(l+1)}$ applied in convolution layer $l$ is characterised by trainable kernel parameters $\Phi^l$. The parameters are learnt by

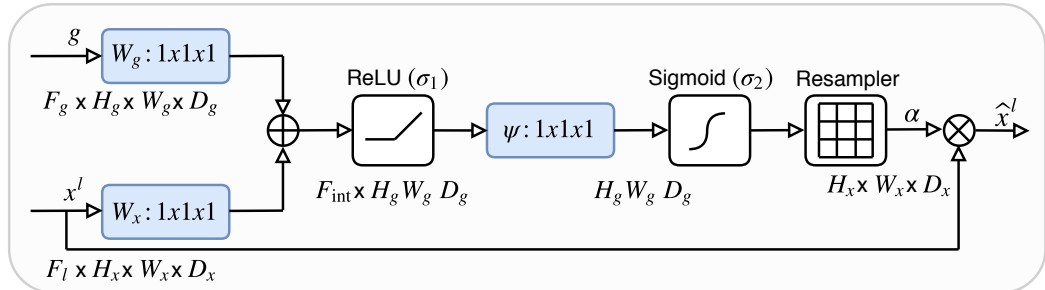

Figure 2: Schematic of the proposed additive attention gate (AG). Input features ($x^l$) are scaled with attention coefficients ($\alpha$) computed in AG. Spatial regions are selected by analysing both the activations and contextual information provided by the gating signal ($g$) which is collected from a coarser scale. Grid resampling of attention coefficients is done using trilinear interpolation.

minimising a training objective, e.g. cross-entropy loss, using stochastic gradient descent (SGD). In this paper, we build our attention model on top of a standard U-Net architecture. U-Nets are commonly used for image segmentation tasks because of their good performance and efficient use of GPU memory. The latter advantage is mainly linked to extraction of image features at multiple image scales. Coarse feature-maps capture contextual information and highlight the category and location of foreground objects. Feature-maps extracted at multiple scales are later merged through skip connections to combine coarse- and fine-level dense predictions as shown in Figure 1.

**Attention Gates for Image Analysis:** To capture a sufficiently large receptive field and thus, semantic contextual information, the feature-map grid is gradually downsampled in standard CNN architectures. In this way, features on the coarse spatial grid level model location and relationship between tissues at global scale. However, it remains difficult to reduce false-positive predictions for small objects that show large shape variability. In order to improve the accuracy, current segmentation frameworks [14, 26, 27] rely on additional preceding object localisation models to simplify the task into separate localisation and subsequent segmentation steps. Here, we demonstrate that the same objective can be achieved by integrating attention gates (AGs) in a standard CNN model. This does not require the training of multiple models and a large number of extra model parameters. In contrast to the localisation model in multi-stage CNNs, AGs progressively suppress feature responses in irrelevant background regions without the requirement to crop a ROI between networks.

Attention coefficients, $\alpha_i \in [0, 1]$, identify salient image regions and prune feature responses to preserve only the activations relevant to the specific task as shown in Figure 3a. The output of AGs is the element-wise multiplication of input feature-maps and attention coefficients: $\hat{x}^l_{i,c} = x^l_{i,c} \cdot \alpha^l_i$. In a default setting, a single scalar attention value is computed for each pixel vector $x^l_i \in \mathbb{R}^{F_l}$ where $F_l$ corresponds to the number of feature-maps in layer $l$. In case of multiple semantic classes, we propose to learn multi-dimensional attention coefficients. This is inspired by [29], where multi-dimensional attention coefficients are used to learn sentence embeddings. Thus, each AG learns to focus on a subset of target structures. As shown in Figure 2, a gating vector $g_i \in \mathbb{R}^{F_g}$ is used for each pixel $i$ to determine focus regions. The gating vector contains contextual information to prune lower-level feature responses as suggested in [32], which uses AGs for natural image classification. We use additive attention [2] to obtain the gating coefficient. Although this is computationally more expensive, it has experimentally shown to achieve higher accuracy than multiplicative attention [19]. Additive attention is formulated as follows:

$$q^l_{att} = \psi^T \left( \sigma_1 \left( W^T_x x^l_i + W^T_g g_i + b_g \right) \right) + b_\psi \tag{1}$$

$$\alpha^l_i = \sigma_2( q^l_{att}(x^l_i, g_i; \Theta_{att})), \tag{2}$$

where $\sigma_2(x_{i,c}) = \frac{1}{1+exp(-x_{i,c})}$ correspond to sigmoid activation function. AG is characterised by a set of parameters $\Theta_{att}$ containing: linear transformations $W_x \in \mathbb{R}^{F_l \times F_{int}}$, $W_g \in \mathbb{R}^{F_g \times F_{int}}$, $\psi \in \mathbb{R}^{F_{int} \times 1}$ and bias terms $b_\psi \in \mathbb{R}$, $b_g \in \mathbb{R}^{F_{int}}$. The linear transformations are computed using channel-wise 1x1x1 convolutions for the input tensors. In other contexts [33], this is referred to as *vector concatenation-based attention*, where the concatenated features $x^l$ and $g$ are linearly mapped to a $\mathbb{R}^{F_{int}}$ dimensional intermediate space. In image captioning [1] and classification [11] tasks, the

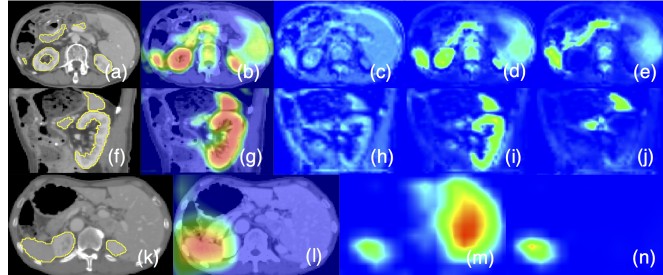 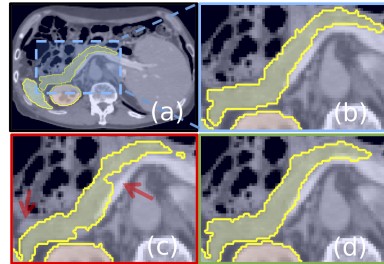

Figure 3(a): From left to right (a-e, f-j): Axial and sagittal views of a 3D abdominal CT scan, attention coefficients, feature activations of a skip connection before and after gating. Similarly, (k-n) visualise the gating on a coarse scale skip connection. The filtered feature activations (d-e, i-j) are collected from multiple AGs, where a subset of organs is selected by each gate. Activations shown in (d-e, i-j) consistently correspond to specific structures across different scans.

Figure 3(b): The ground-truth pancreas segmentation (a) is highlighted in blue (b). Similarly, U-Net model prediction (c) and the predictions obtained with Attention U-Net (d) are shown. The missed dense predictions by U-Net are highlighted with red arrows.

softmax activation function is used to normalise the attention coefficients ($\sigma_2$); however, sequential use of softmax yields sparser activations at the output. For this reason, we choose a sigmoid activation function. This results experimentally in better training convergence for the AG parameters. In contrast to [11] we propose a grid-attention technique. In this case, gating signal is not a global single vector for all image pixels but a grid signal conditioned to image spatial information. More importantly, the gating signal for each skip connection aggregates information from multiple imaging scales, as shown in Figure 1, which increases the grid-resolution of the query signal and achieve better performance. Lastly, we would like to note that AG parameters can be trained with the standard back-propagation updates without a need for sampling based update methods used in hard-attention [21].

**Attention Gates in U-Net Model:** The proposed AGs are incorporated into the standard U-Net architecture to highlight salient features that are passed through the skip connections, see Figure 1. Information extracted from coarse scale is used in gating to disambiguate irrelevant and noisy responses in skip connections. This is performed right before the concatenation operation to merge only relevant activations. Additionally, AGs filter the neuron activations during the forward pass as well as during the backward pass. Gradients originating from background regions are down weighted during the backward pass. This allows model parameters in shallower layers to be updated mostly based on spatial regions that are relevant to a given task. The update rule for convolution parameters in layer $l-1$ can be formulated as follows:

$$\frac{\partial(\hat{x}_i^l)}{\partial\left(\Phi^{l-1}\right)} = \frac{\partial\left(\alpha_i^l\, f(x_i^{l-1};\Phi^{l-1})\right)}{\partial\left(\Phi^{l-1}\right)} = \alpha_i^l\,\frac{\partial(f(x_i^{l-1};\Phi^{l-1}))}{\partial\left(\Phi^{l-1}\right)} + \frac{\partial(\alpha_i^l)}{\partial\left(\Phi^{l-1}\right)}\,x_i^l \tag{3}$$

The first gradient term on the right-hand side is scaled with $\alpha_i^l$. In case of multi-dimensional AGs, $\alpha_i^l$ corresponds to a vector at each grid scale. In each sub-AG, complementary information is extracted and fused to define the output of skip connection. To reduce the number of trainable parameters and computational complexity of AGs, the linear transformations are performed without any spatial support (1x1x1 convolutions) and input feature-maps are downsampled to the resolution of gating signal, similar to non-local blocks [33]. The corresponding linear transformations decouple the feature-maps and map them to lower dimensional space for the gating operation. As suggested in [11], low-level feature-maps, i.e. the first skip connections, are not used in the gating function since they do not represent the input data in a high dimensional space. We use deep-supervision [16] to force the intermediate feature-maps to be semantically discriminative at each image scale. This helps to ensure that attention units, at different scales, have an ability to influence the responses to a large range of image foreground content. We therefore prevent dense predictions from being reconstructed from small subsets of skip connections.

## 3   Experiments and Results

The proposed AG model is modular and independent of application type; as such it can be easily adapted for classification and regression tasks. To demonstrate its applicability to image segmentation,

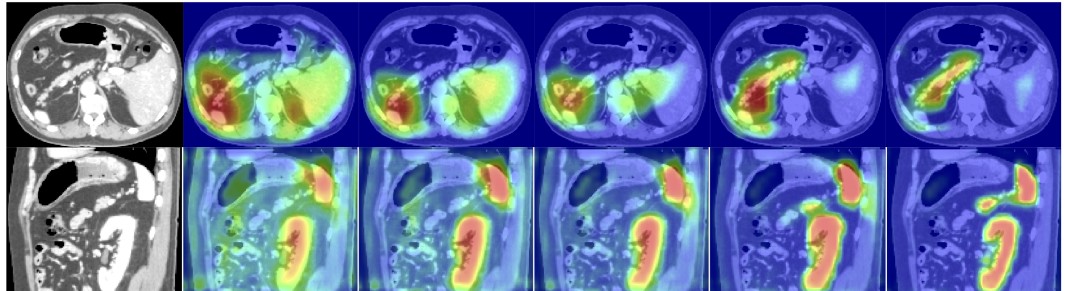

Figure 4: The figure shows the attention coefficients ($\alpha^{l_{s_2}}$, $\alpha^{l_{s_3}}$) across different training epochs (3, 6, 10, 60, 150). The images are extracted from sagittal and axial planes of a 3D abdominal CT scan from the testing dataset. The model gradually learns to focus on the pancreas, kidney, and spleen.

we evaluate the Attention U-Net model on a challenging abdominal CT multi-label segmentation problem. In particular, pancreas boundary delineation is a difficult task due to shape-variability and poor tissue contrast. Our model is compared against the standard 3D U-Net in terms of segmentation performance, model capacity, computation time, and memory requirements.

**Evaluation Datasets:** For the experiments, two different CT abdominal datasets are used: (I) 150 abdominal 3D CT scans acquired from patients diagnosed with gastric cancer ($CT$-150). In all images, the pancreas, liver, and spleen boundaries were semi-automatically delineated by three trained researchers and manually verified by a clinician. The same dataset is used in [27] to benchmark the U-Net model in pancreas segmentation. (II) The second dataset[1] ($CT$-82) consists of 82 contrast enhanced 3D CT scans with pancreas manual annotations performed slice-by-slice. This dataset (NIH-TCIA) [25] is publicly available and commonly used to benchmark CT pancreas segmentation frameworks. The images from both datasets are downsampled to isotropic 2.00 mm resolution due to the large image size and hardware memory limitations.

**Implementation Details:** In contrast to the state-of-the-art CNN segmentation frameworks [4, 26], we propose a 3D-model to capture sufficient semantic context. Gradient updates are computed using small batch sizes of 2 to 4 samples. For larger networks, gradient averaging is used over multiple forward and backward passes. All models are trained using the Adam optimiser [15], batch-normalisation, deep-supervision [16], and standard data-augmentation techniques (affine transformations, axial flips, random crops). Intensity values are linearly scaled to obtain a normal distribution $N(0, 1)$. The models are trained using the Sorensen-Dice loss [20] defined over all semantic classes, which is experimentally shown to be less sensitive to class imbalance. Gating parameters are initialised so that attention gates pass through feature vectors at all spatial locations. Moreover, we do not require multiple training stages as in hard-attention based approaches therefore simplifying the training procedure. Our implementation using PyTorch is publicly available[2].

**Attention Map Analysis:** The attention coefficients obtained from test images are visualised with respect to training epochs (see Figure 4). We commonly observe that AGs initially have a uniform distribution and pass features at all locations. This is gradually updated and localised towards the targeted organ boundaries. Additionally, at coarser scales AGs provide a rough outline of organs which are gradually refined at finer resolutions. Moreover, by training multiple AGs at each image scale, we observe that each AG learns to focus on a particular subset of organs.

**Segmentation Experiments**: The proposed Attention U-Net model is benchmarked against the standard U-Net on multi-class abdominal CT segmentation. We use $CT$-150 dataset for both training (120) and testing (30). The corresponding Dice scores (DSC) and surface distances (S2S) are given in Table 1. The results on pancreas predictions demonstrate that attention gates (AGs) increase recall values ($p = .005$) by improving the model's expression power as it relies on AGs to localise foreground pixels. The difference between predictions obtained with these two models are qualitatively compared in Figure 3b. In the second experiment, the same models are trained with fewer training images (30) to show that the performance improvement is consistent and significant for different sizes of training data ($p = .01$). For both approaches, we observe a performance drop on

---

[1]https://wiki.cancerimagingarchive.net/display/Public/Pancreas-CT
[2]https://github.com/ozan-oktay/Attention-Gated-Networks

Table 1: Multi-class CT abdominal segmentation results obtained on the $CT$-150 dataset: The results are reported in terms of Dice score (DSC) and mesh surface to surface distances (S2S). These distances are reported only for the pancreas segmentations. The proposed Attention U-Net model is benchmarked against the standard U-Net model for different training and testing splits. Inference time (forward pass) of the models are computed for input tensor of size $160 \times 160 \times 96$. Statistically significant results are highlighted in bold font.

| Method (Train/Test Split) | U-Net (120/30) | Att U-Net (120/30) | U-Net (30/120) | Att U-Net (30/120) |
|---|---|---|---|---|
| Pancreas DSC | 0.814±0.116 | **0.840±0.087** | 0.741±0.137 | **0.767±0.132** |
| Pancreas Precision | 0.848±0.110 | 0.849±0.098 | 0.789±0.176 | **0.794±0.150** |
| Pancreas Recall | 0.806±0.126 | **0.841±0.092** | 0.743±0.179 | **0.762±0.145** |
| Pancreas S2S Dist (mm) | 2.358±1.464 | **1.920±1.284** | 3.765±3.452 | 3.507±3.814 |
| Spleen DSC | 0.962±0.013 | 0.965±0.013 | 0.935±0.095 | **0.943±0.092** |
| Kidney DSC | 0.963±0.013 | 0.964±0.016 | 0.951±0.019 | 0.954±0.021 |
| Number of Params | 5.88 M | 6.40 M | 5.88 M | 6.40 M |
| Inference Time | 0.167 s | 0.179 s | 0.167 s | 0.179 s |

Table 2: Segmentation experiments on $CT$-150 dataset are repeated with higher capacity U-Net models to demonstrate the efficieny of the attention models with similar or less network capacity. The additional filters in the U-Net model are distributed uniformly across all the layers.

| Method | Panc. DSC | Panc. Precision | Panc. Recall | S2S Dist (mm) | # of Pars | Run Time |
|---|---|---|---|---|---|---|
| U-Net (120/30) | 0.821±.119 | 0.849±.111 | 0.814±.125 | 2.383±1.918 | 6.44 M | 0.191 s |
| U-Net (120/30) | 0.825±.104 | 0.861±.082 | 0.807±.121 | 2.202±1.144 | 10.40 M | 0.222 s |

Table 3: Pancreas segmentation results obtained on the TCIA Pancreas-CT Dataset [25]. The dataset contains in total 82 scans which are split into training (61) and testing (21) sets. The corresponding results are obtained before (BFT) and after fine tuning (AFT) and also training the models from scratch (SCR). Statistically significant results are highlighted in bold font.

| | Method | Dice Score | Precision | Recall | S2S Dist (mm) |
|---|---|---|---|---|---|
| BFT | U-Net [24] | 0.690±0.132 | 0.680±0.109 | 0.733±0.190 | 6.389±3.900 |
| | Attention U-Net | **0.712±0.110** | 0.693±0.115 | **0.751±0.149** | **5.251±2.551** |
| AFT | U-Net [24] | 0.820±0.043 | 0.824±0.070 | 0.828±0.064 | 2.464±0.529 |
| | Attention U-Net | **0.831±0.038** | 0.825±0.073 | **0.840±0.053** | **2.305±0.568** |
| SCR | U-Net [24] | 0.815±0.068 | 0.815±0.105 | 0.826±0.062 | 2.576±1.180 |
| | Attention U-Net | 0.821±0.057 | 0.815±0.093 | **0.835±0.057** | **2.333±0.856** |

spleen DSC as the training size is reduced. The drop is less significant with the proposed framework. For kidney segmentation, the models achieve similar accuracy since the tissue contrast is higher.

In Table 1, we also report the number of trainable parameters for both models. We observe that by adding 8% extra capacity to the standard U-Net, the performance can be improved by 2-3% in terms of DSC. For a fair comparison, we also train higher capacity U-Net models and compare against the proposed model with smaller network size. The results shown in Table 2 demonstrate that the addition of AGs contributes more than simply increasing model capacity (uniformly) across all layers of the network ($p = .007$). Therefore, additional capacity should be used for AGs to localise tissues, in cases when AGs are used to reduce the redundancy of training multiple, individual models.

**Comparison to State-of-the-Art:** The proposed architecture is evaluated on the public TCIA CT Pancreas benchmark to compare its performance with state-of-the-art methods. Initially, the models trained on $CT$-150 dataset are directly applied to $CT$-82 dataset to observe the applicability of the two models on different datasets. The corresponding results (BFT) are given in Table 3. U-Net model outperforms traditional atlas techniques [34] although it was trained on a disjoint dataset. Moreover, the attention model performs consistently better in pancreas segmentation across different datasets. These models are later fine-tuned (AFT) on a subset of TCIA dataset (61 train, 21 test). The output nodes corresponding to spleen and kidney are excluded from the output softmax computation, and the gradient updates are computed only for the background and pancreas labels. The results in Table 3 and

Table 4: State-of-the-art CT pancreas segmentation methods that are based on single and multiple CNN models. The listed segmentation frameworks are evaluated on the same public benchmark ($CT$-82) using different number of training and testing images. Similarly, the FCN approach proposed in [27] is benchmarked on $CT$-150 although it is trained on an external dataset (Ext).

| Method | Dataset | Pancreas DSC | Train/Test | # Folds |
|---|---|---|---|---|
| Hierarchical 3D FCN [27] | $CT$-150 | $82.2 \pm 10.2$ | Ext/150 | - |
| Dense-Dilated FCN [6] | $CT$-82 & Synapse[3] | $66.0 \pm 10.0$ | 63/9 | 5-CV |
| 2D U-Net [8] | $CT$-82 | $75.7 \pm 9.0$ | 66/16 | 5-CV |
| Holistically Nested 2D FCN Stage-1[26] | $CT$-82 | $76.8 \pm 11.1$ | 62/20 | 4-CV |
| Holistically Nested 2D FCN Stage-2[26] | $CT$-82 | $81.2 \pm 7.3$ | 62/20 | 4-CV |
| 2D FCN [4] | $CT$-82 | $80.3 \pm 9.0$ | 62/20 | 4-CV |
| 2D FCN + Recurrent Network [4] | $CT$-82 | $82.3 \pm 6.7$ | 62/20 | 4-CV |
| Single Model 2D FCN [38] | $CT$-82 | $75.7 \pm 10.5$ | 62/20 | 4-CV |
| Multi-Model 2D FCN [38] | $CT$-82 | $82.2 \pm 5.7$ | 62/20 | 4-CV |

4 show improved performance compared to concatenated multi-model CNN approaches [4, 26, 38] due to additional training data and richer semantic information (e.g. spleen labels). Additionally, we trained the two models from scratch (SCR) with 61 training images randomly selected from the $CT$-82 dataset. Similar to the results on $CT$-150 dataset, AGs improve the segmentation accuracy and lower the surface distances ($p = .03$) due to increased recall rate of pancreas pixels ($p = .09$).

Results from state-of-the-art CT pancreas segmentation models are summarised in Table 4 for comparison purposes. Since the models are trained on the same training dataset, this comparison gives an insight on how the attention model compares to the relevant literature. It is important to note that, post-processing (e.g. conditional random field) is not utilised in our framework as the experiments mainly focus on quantification of performance improvement brought by AGs in an isolated setting. Similarly, residual and dense connections can be used as in [6] in conjunction with AGs to improve the segmentation results. In that regard, our 3D Attention U-Net model performs similar to the state-of-the-art, despite the input images are downsampled to lower resolution. More importantly, our approach significantly improves the results compared to single-model based segmentation frameworks (see Table 4). We do not require multiple CNN models to localise and segment object boundaries. Lastly, we performed 5-fold cross-validation on the $CT$-82 dataset using the Attention U-Net for a better comparison, which achieved $81.48 \pm 6.23$ DSC for pancreas labels.

## 4    Discussion and Conclusion

In this paper, we presented a novel attention gate model applied to medical image segmentation. Our approach eliminates the necessity of applying an external object localisation model. The proposed approach is generic and modular as such it can be easily applied to image classification and regression problems as in the examples of natural image analysis and machine translation. Experimental results demonstrate that the proposed AGs are highly beneficial for tissue/organ identification and localisation. This is particularly true for variable small size organs such as the pancreas, and similar behaviour is expected for global classification tasks.

Training behaviour of the AGs can benefit from transfer learning and multi-stage training schemes. For instance, pre-trained U-Net weights can be used to initialise the attention network, and gates can be trained accordingly in the fine-tuning stage. Similarly, there is a vast body of literature in machine learning exploring different gating architectures. For example, highway networks [7] make use of residual connections around the gate block to allow better gradient backpropagation and slightly softer attention mechanisms. Although our experiments with residual connections have not provided any significant performance improvement, future research will focus on this aspect to obtain a better training behaviour. Lastly, we note that with the advent of improved GPU computation power and memory, larger capacity 3D models can be trained with larger batch sizes without the need for image downsampling. In this way, we would not need to utilise ad-hoc post-processing techniques to further improve the state-of-the-art results. Similarly, the performance of Attention U-Net can be further enhanced by utilising fine resolution input batches without additional heuristics. Lastly, we would like to thank to Salim Arslan and Dan Busbridge for their helpful comments on this work.

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
