# OpenReview forum: "Attention U-Net: Learning Where to Look for the Pancreas"
_MIDL.amsterdam/2018/Conference — MIDL 2018 Oral_

### Review · AnonReviewer1 · 2018-05-04
**convoluted presentation of a simple idea, experimental validation is rather weak**

**Rating:** 2
**Confidence:** 2

**Review:**

This paper tackles the important problem of organ segmentation, by introducing a gated attention mechanism to  filter the features skipped from the downsampling to the upsampling path.

pros
+ simple pipeline

cons
- very convoluted presentation of a very simple idea
- experimental validation is weak

The presentation of the paper could be significantly improved, the proposed approach is very simple but it is explained in a very convoluted way. Notation between text and figures does not always match.

In section 2, the FCN description should be reviewed. x^l is introduced as a feature map (representation), obtained after processing a layer. Then, x^ l_{c,i} is introduced as a feature map. I understand the superscript l refers to the layer, but what are the subscripts c and i? It seems that one subscript, to refer to a particular channel in the feature map, would suffice in this case. Subscript i is defined after a formula which omits the subscript. Moreover, introducing the operations performed by a standard convolutional layer followed by ReLU activation might not be necessary.

The proposed model is based on the U-Net architecture, which was first introduced in a 2D setting. The referenced FCN was also introduced in a 2D setting. Therefore, it would be beneficial to mention that the U-Net architecture used in this paper is 3D in Section 2 (this information is only available in Section 3).

W_x and W_g, which are 3D convolutional kernels, are defined in the text as 2D matrices (but should be 5D?). There is a lot of redundancy w.r.t. the parameter notation, which keeps changing throughout the paper.

Figure 2 illustrates the proposed gated attention mechanism. When introducing Figure 2, it is not clear that coarser representations are used as gating inputs. Moreover, the Figure is misleading: the proper dimensions of W_x and W_g should be included, the downsampling operations performed after convolving x^ l with W_x should be shown (otherwise it seems the model is trying to sum two tensors of different dimensions).

When presenting "Attention gates in U-Net Model", the l superscript of alpha disappears.

Given the increase in number of parameters led by the multi-dimensional attention coefficients, analyzing the impact of using a single coefficient per position and comparing the results with the multi-dimensional ones would provide additional insights.

Since deep supervision is used when adding gated attention, it would be important to provide results without the deep supervision (or adding deep supervision to U-Net) for fair comparison.

The presentation of Figure 3(b) could be improved. Referring to the different segmentation as (a, b, c, d) instead of using colors around the figures would be beneficial.

Is dataset 1 (CT-150) publicly available? (there is no link to it).

Is Table 1 presented in 2 lines because it does not fit the page? Why not present it the other way around (8 rows for results, number of parameters and time) and 4 columns (for the 4 models, you could include the split in the model's name for clarity)?

How many folds are used to report results in Table 1? Why not use the same splits as in [25]?

Table 2 could be merged with Table 1. Is the performance improvement of the gated attention w.r.t. U-Net (6.44M parameters) significant? Which model should we be comparing to 120/30 or 30/120?

Training details are missing. What optimizer is used? Do you apply dropout, batch norm, ...?

SCR should appear next to scratch in the text.

It seems that the results reported on Table 3 do not follow the same splits as in Table 4(61/21 vs 62/20 for the majority of the methods). It would be important to report results on the same splits (4-CV, since most of the models do that), and merge Table 3 and Table 4 to ease the comparison among different approaches. Final results on the method should be included in the Table as well (not only the text, and with 5-CV, making it hard to assess how the method performs compared to others).

The authors claim that their method can achieve similar results than state-of-the-art while working with downsampled volumes. However, it seems that most (or all?) of the state-of-the-art models presented in the table operate on 2D slices.

Some missing references:

Dice loss:
[a] https://arxiv.org/abs/1608.04117
[b] https://arxiv.org/abs/1606.04797

Attention Gates for image analysis section reviews attention literature, I suggest removing "image analysis part". A couple of additional papers could be included in this section:
[c] https://arxiv.org/abs/1706.03762
[d] https://arxiv.org/abs/1710.10903

Pytorch:
[e] https://openreview.net/forum?id=BJJsrmfCZ

**Special Issue:**

No

---

### Review · AnonReviewer3 · 2018-05-09
**Attention U-Net: Learning Where to Look for the Pancreas**

**Rating:** 5
**Confidence:** 3

**Review:**

The paper presents a simple yet effective gate block, which can be plugged in to implement an attention mechanisms in existing convolutional neural network architectures.
The U-Net architecture is considered in this paper, applied to the specific problem of organs segmentation in thorax CT images.
Results show the effectiveness of the proposed attention gate, which allows to improve segmentation performance.
Authors also show experimentally that improvement is not simply due to increased capacity of the network, but to the way the attention gate works.
The paper is very well written, the proposed method is clearly presented and the validation is thorough.

My only comment is about the network architecture and the application that was chosen to showcase the added value of such an attention gate.
In figure 3 and 4, the authors show that the attention mechanism allows the network to focus on some regions of the CT scan, which correspond to organs.
Although this is interesting, it would be even more interesting to explore attention mechanisms in image classification problems, where the target object is not manually defined, and the network should learn to attend to regions that are linked to the image-level label.


**Special Issue:**

Definitely

---

### Review · AnonReviewer2 · 2018-05-10
**Interesting paper, well described, evaluated on public data and with source code available. Excellent!**

**Rating:** 4
**Confidence:** 2

**Review:**

The authors present an interesting method to add a soft attention mechanism to a U-net network structure. They show good performance, especially when compared to other single network solutions. Overall I liked the paper, I think the topic is interesting and relevant for a number of applications in medical imaging.

Pros:
- well written
- method clearly described
- evaluated on public data
- source code for the method made available
- interesting and generally applicable attention method

Cons:
- The performance, while very good for a single network solution, does not blow other methods with multiple networks out of the water

Detailed remarks:

Page 8: I was surprised by the provided standard deviation in the cross validation experiment (0.062). That is so much lower than those from the other methods in table 4 that I am wondering if the authors used a different method to calculate it than  the other methods in table 4.

**Special Issue:**

Yes

---

### Comment · ~Bram_van_Ginneken1 · 2018-05-18
**Selection for longlist for special issue Medical Image Analysis**

Dear authors,

Congratulations on your acceptance to MIDL! We have selected your paper on the longlist for the Medical Image Analysis Special Issue. Please read this page:
https://midl.amsterdam/special-issue-in-medical-image-analysis/
Please answer the three questions that are listed on that page about your interest in submitting to the special issue, potential overlap with other publications, and related publications.

You can post your answer here directly below on openreview.net, or mail me directly at bram.vanginneken@radboudumc.nl.

Best regards, Bram

---

### Decision · Program_Chairs · 2018-05-15
**Paper29 Acceptance Decision**

Oral